# Investigation into Pigmentation Behaviors and Mechanism of Pigment Yellow 180 in Different Solvents

**Kairu Ye** [1], **Yan Yang** [2], **Haishuo Chen** [2], **Jiatong Wu** [2], **Hongyuan Wei** [1] **and Leping Dang** [1,*]

[1]  School of Chemical Engineering and Technology, Tianjin University, Tianjin 300072, China; 2020207058@tju.edu.cn (K.Y.); david.wei@tju.edu.cn (H.W.)

[2]  Anshan Hifichem Co., Ltd., No. 8, 1st Bao An Road, Teng Ao Industrial Park, Anshan 114225, China; yangyan@hifichem.com (Y.Y.); chenhaishuo@hifichem.com (H.C.); wujiatong@hifichem.com (J.W.)

[*]  Correspondence: dangleping@tju.edu.cn

**Abstract:** To achieve the target colors, pigmentation (post-processing) in solvents is a key process in making Pigment Yellow 180 (PY180), a bis azo pigment. In this work, the solvent effect on the pigmentation behavior of PY180 was studied based on the Hansen solubility parameters (HSPs) and the molecular polarity index (MPI) method. First, the samples were characterized using FTIR, XRD, and TEM, and the colorimetric analysis was performed using the CIE L*a*b* color space model. It was found that the color hues obtained in ten solvents are different, with the overall color variation from reddish–yellow to greenish–yellow. Further characterization confirmed that the crystallinity and particle size increase of PY180 during the pigmentation mainly account for the variation of the chromaticity. Then, HSPs were introduced to understand how suspension behavior affects the dissolution–reprecipitation process. It shows that high-quality pigments can be obtained from solvents generally with low HSP differences ($\Delta\delta$) between the solvents and PY180. To compensate for the inaccurate prediction of the HSPs method, MPI was used to value the influence of solvent molecular polarity. The results show that among solvents with similar solubility parameters to PY180, the stronger the molecular polarity index (MPI) of the solvent, the greater the color variation of the pigments. Meanwhile, different solvents influence the crystallization behavior of the low soluble system, which was supplemented by the above study.

**Keywords:** pigment; solvent effect; crystallization

## 1. Introduction

Pigments are used almost everywhere in people's daily life. Organic pigments are synthesized to replace some inorganic pigments because of their low toxicity, bright color, and high coloring power [1,2]. Among the synthetic organic pigments, azo pigments are gradually taking over the field of red and yellow pigments; more than 80% of yellow and 65% of red pigments have azo-linkages [3]. However, pigments generally do not achieve the target color after synthesis and usually require some post-treatment methods such as heating the pigment in water or organic solvents (suspensions), solvent grinding or ball milling [4,5], supercritical antisolvent process (CO$_2$) [6], and calcination [7]. These post-treatments are generally used for achieving crystallinity and increasing the crystallinity to obtain the desired color, hue, color strength, and hiding power.

C.I. Pigment Yellow 180 (PY180, 2,2′-[ethylenebis(oxyphenyl-2,1-eneazo)]*bis*[N-(2,3-dihydro2-oxo-1*H*-benzimidazol-5-yl)-3-oxobutyramide), the only disazo-benzimidazolone pigment, is a greenish to medium yellow pigment with strong solvent resistance, lightfastness (step 6–7 on the Blue Scale), heat stability, and good tinctorial strength, mainly used in printing ink and the plastic industry [8,9]. Azo pigments are synthesized at a low temperature, leading to a rapid precipitation of amorphous particles and small sizes of crystals. Therefore, they need to be further treated. The common post-treatment method is

the solvent thermal method, where a suspension is prepared in a solvent at a high temperature and pressure. The selection of an appropriate organic solvent allows the dissolution of byproducts to remove the impurities and complete the recrystallization to increase the particle size.

Researchers have reported the properties of the solvent itself, such as polarity, surface tension, dielectric constant, geometry, pH, and viscosity, which have been investigated to explain the mechanisms of material crystal growth [10–15]. At the same time, little literature has been reported on the pattern and mechanism of solvent effects on the pigmentation process of pigments, which are generally low in solubility in solvents. To the best of our knowledge, in the field of pigments, Hansen solubility parameters (HSPs) are directly related to the suspension behavior and relative settling time in solvent mixtures and their affinity for other components and have been successfully used to select the appropriate solvent for coating materials [16–18]. Recently, Mao et al. reported an optimal solvent selection method based on solubility parameters for triggering Ostwald ripening [19]. Several studies have investigated the compatibility of two molecules through HSPs to explore cocrystal formation [20]. Inspired by these studies, we used HSPs as a tool to guide solvent selection for pigmentation. In addition to this, there are many published investigations that have found a relation between solvent–solute interaction with nucleation and crystal growth as determined by molecular simulations [21–24]. The mechanism of intrinsic interaction could be complemented by discussing the electronic and geometrical structure of the molecule.

This study explored the trends in pigment chromaticity changes and the variation in the chemical and physical structure of PY180 particles during the pigmentation in different solvents, which will provide a general understanding of the relation between pigments and solvents. Then, by investigating the dispersion and HSPs of pigments, we can gain insights into the particle–solvent interaction and elucidate how it influences the recrystallization process. Furthermore, to complement the understanding of the intrinsic interaction from a molecular level, we discuss the electronic and geometrical structure of the pigment molecules. By analyzing these aspects, we aim to provide some guidance on the solvent–pigment interaction mechanism and solvent selection for the pigmentation process.

## 2. Materials and Methods

### 2.1. Materials

PY180 raw material was obtained from Anshan Hifichem Co., Ltd. (Anshan, China). Methanol (MeOH, 99.9%), ethanol (EtOH, 99.9%), 1-propanol (PrOH, 99.5%), and 1-butanol (BuOH, 99.5%) were obtained from Kermel. *N*, *N*-dimethylformamide (DMF, 99%), *N*-methyl-2-pyrrolidone (NMP, 99%), dimethyl sulfoxide (DMSO, 99%), acetonitrile (CAN, 99%), and propylene glycol monomethyl ether acetate (PGME with 50 ppm BHT) were purchased from Aladdin. The chemicals were used as received without any further treatment.

### 2.2. Sample Preparation

**PY180 pigmentation:** 12 g of PY180 raw material (PY180-raw) was dispersed in 300 mL of methanol (MeOH) solvent and stirred continuously for 2 h to form a suspension. The PY180 suspension was placed in an autoclave, which was heated up to 140 °C within 1.5 h and held at 140 °C for 10 h. The samples were collected at intervals over the holding time (0 h, 1 h, 2 h, 4 h, 6 h, 8 h, 10 h). The autoclave needs to be cooled down to the room temperature at each collection. The collected particles were filtered and washed with distilled water and then dried under a vacuum condition at 60 °C overnight until the solvent was completely evaporated so that PY180 pigment was obtained. In addition, experiments were also conducted in different solvents (EtOH, PrOH, BuOH, hexane, DMSO, DMF, NMP, $H_2O$) under the same pigmentation conditions. The collected samples are named PY180-solvent, corresponding to the solvent used.

**Chromaticity analysis:** PY180-solvent (0.05 g) and PGMEA (1 mL) were mixed, and PY180-PGMEA suspension was obtained by sonication (40 Hz) for 30 min. Top-PY180 glass sheets were fabricated by volatilizing PGMEA at room temperature by placing 300 µm drops of PY180-PGMEA on glass sheets (24 mm × 24 mm × 1 mm). Three sets of glass sheets were prepared for each sample for subsequent measurements of L*, a*, and b* values.

Color variation was calculated using the equation below:

$$\Delta E = \sqrt{(\Delta L^*)^2 + (\Delta a^*)^2 + (\Delta b^*)^2}$$

L*a*b* values means bright, red(+)/green(−), and blue(−)/yellow(+), respectively. ΔE was regarded as the total color differences between raw samples and pigmentation samples.

**Solubility parameter experiments:** The dispersion of PY180 in solvents was determined using the solubility parameters. A small amount of PY180-raw was dispersed in different solvents with known solubility parameters, and the suspension was prepared by sonication (1 h, 40 kHz) followed by centrifugation at 600 rpm for 1 h. The upper 2/3 of the clear solution was obtained as PY180-raw/solvent by pipetting. Subsequently, UV absorption spectra of supernatants were measured in the 300–800 nm (10 mm optical path length of the sample, Lambda 750 s UV spectrophotometer, Perkin Elmer (Shelton, CT, USA), which were diluted 2-fold in advance.

**Computational methods**: Geometry optimization was performed using the Gaussian 09 program [25]. The PY180 molecular geometry was optimized under B3LYP/31G(d,p) [26] level with empirical dispersion correction [27].

Quantitative molecular surface analysis was carried out on the wavefunction at the B3LYP/Def2-QZVP [28] level using Multiwfn 3.8 dev [29]. The analysis consisted of the potential energy surface (ESP), vdW potential, and MPI [30]. Surface electrostatic potential maps of the PY180 molecule were generated using files exported by Multiwfn and then rendered in Visual Molecular Dynamics (VMD 1.9.3) software [31].

### 2.3. Characterization

Infrared (IR) spectroscopic analysis was conducted using a Bruker Vertex 70 v instrument in the range of 400–4000 cm$^{-1}$, and Raman spectra were performed using an inVia Raman spectroscopy instrument (532 nm laser) in the range of 200–2000 cm$^{-1}$. Powder X-ray diffraction (XRD) was carried out using a Smartlab X-ray diffractometer equipped with Cu Kα radiation at a scan rate of 5°/min, a 2θ angular range of 5–50°, and a step size of 0.02° for crystal structure characterization. For microstructural characterization of the pigment, a Regulus 8100 scanning electron microscope (SEM) and JEM 2100F transmission electron microscope (TEM) were used. The diffuse reflectance spectroscopy of prepared powders (300–800 nm) and UV spectrophotometric measurements of the dispersed solvents (300–800 nm) were carried out using a Lambda 750 s UV spectrophotometer. To record the L*a*b* values on glass sheets covered with pigments, a CR-10 spectrophotometer was used.

## 3. Results and Analysis

### 3.1. Changing Trends of Chromaticity

Ten solvents were selected as the pigmentation media to evaluate the effect of the solvents on pigment PY180. The solvothermal conditions, such as the pigmentation time (10 h), the holding temperature (140 °C), and the stirring power (200 rpm), remained constant.

The CIE L*a*b* color space model was used to analyze the chromaticity of the samples. Table 1 shows the specific L*a*b* values describing the differences produced by the solvent on the color change of the pigment. After treating with different solvents, the pigments exhibited a tendency towards higher L* and b* values and lower a* values. To clearly represent the color of the pigment after pigmentation, Figure 1a shows the color block of PY180-solvent, which is based on the L*a*b* values measured using a spectrophotometer.

It was found that the color hues obtained in different solvents are different. The color is dullest in *n*-pentane and H$_2$O and brightest in DMSO, with an overall color variation from reddish–yellow to greenish–yellow.

**Table 1.** L*a*b* color analysis for PY180.

|  | L* | a* | b* | ΔL* | Δa* | Δb* | ΔE |
|---|---|---|---|---|---|---|---|
| PY180-raw | 74.00 | 18.72 | 98.70 | 0.00 | 0.00 | 0.00 | 0.00 |
| PY180-Hexane | 77.08 | 17.54 | 101.56 | 3.08 | −1.18 | 2.86 | 4.37 |
| PY180-BuOH | 78.60 | 12.40 | 100.37 | 4.60 | −6.32 | 1.67 | 7.99 |
| PY180-PrOH | 79.79 | 11.00 | 100.20 | 5.79 | −7.72 | 1.50 | 9.77 |
| PY180-EtOH | 82.30 | 9.50 | 102.34 | 8.30 | −9.22 | 3.64 | 12.93 |
| PY180-MeOH | 83.52 | 8.18 | 103.67 | 9.52 | −10.54 | 4.97 | 15.05 |
| PY180-NMP | 86.02 | 6.66 | 105.52 | 12.02 | -12.06 | 6.82 | 18.35 |
| PY180-DMF | 86.97 | 6.01 | 105.30 | 12.97 | -12.71 | 6.60 | 19.32 |
| PY180-DMSO | 87.39 | 5.04 | 105.07 | 13.39 | -13.68 | 6.37 | 20.17 |
| PY180-H$_2$O | 75.04 | 12.36 | 95.98 | 1.04 | -6.36 | -2.72 | 6.99 |

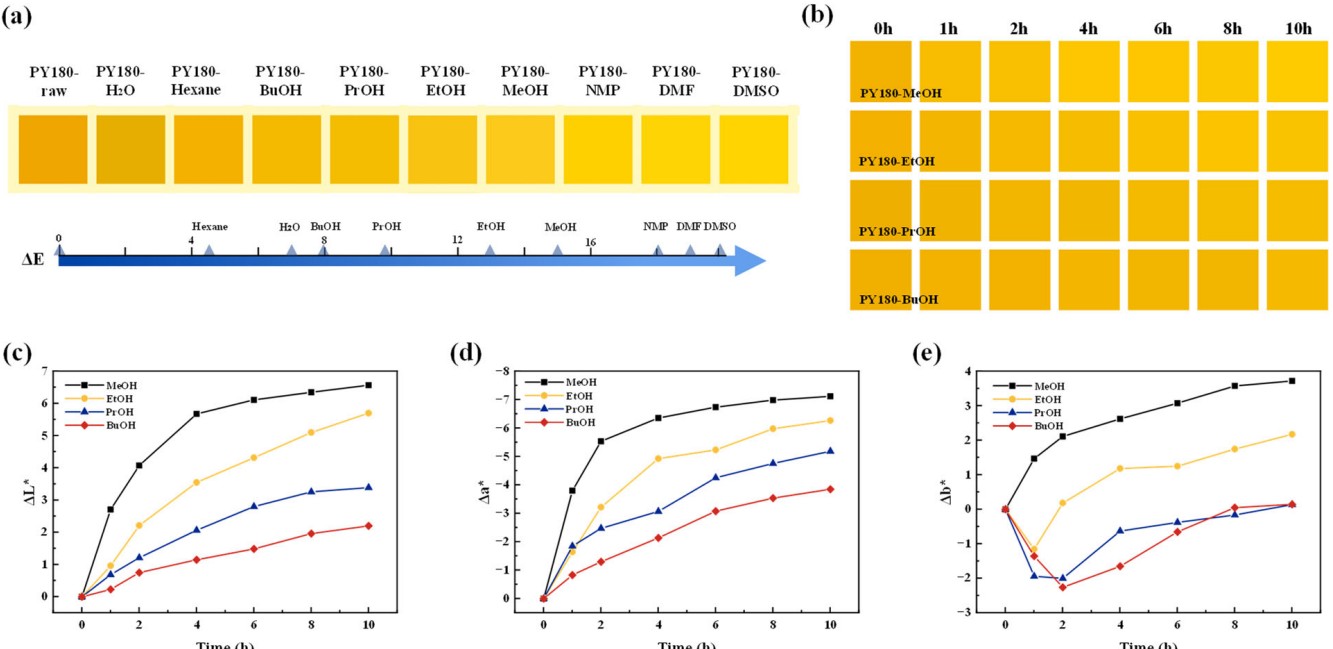

**Figure 1.** Color block of (**a**) PY180-solvent and (**b**) PY180 treated with four alcohol solvents at different times. L*a*b* color analysis of PY180: (**c**) ΔL* (lightness), (**d**) Δa*, (**e**) Δb* vs. treatment time.

Alcohol solvents are the most prevalent in the pigmentation of organic pigments. In addition, regulating the pigmentation time allows us to study the trend of pigment change in the solvent. The variation pattern of PY180 with time and alcohol solvent is shown in Figure 1b (L*a*b* values shown in Table S1), allowing more visual observation of the pigment color change. As the pigmentation time increases, it shows a green shift, and the color gradually becomes brighter. The L*a*b* value of methanol showed the most obvious change. The graph in Figure 1c–e illustrates a high rate of change in the first 2 h of treatment but gradually decreases with prolonging the pigmentation time. The UV diffuse reflection spectra (Figure S1) show that samples treated in alcohol solvents had reflections in the red wavelength (770–622 nm), orange wavelength (622–597 nm), yellow wavelength (597–577 nm), and green wavelength (577–492 nm) bands. After alcohol solvent treatment, the reflectance peak shifted to lower wavelengths, and the reflectance in the yellow and green wavelengths (500–600 nm) increased, indicating an increase in the yellow–green hue

of the color. The overall increase in reflectance corresponds to an increase in brightness, as represented by the L\*a\*b\* value.

### 3.2. Chemical and Physical Structure Characterization

Many factors affect color, such as chemical structure, crystallinity, crystal shape, particle size, and morphology. The molecular structures of PY180 before and after the pigmentation were first assessed using Raman and IR spectra. Raman is associated with molecular polarizability, while IR involves dipole changes in the molecule.

Figure 2 shows the IR and Raman spectra for PY180. The absorption peaks of the IR spectra before pigmentation were consistent with the IR of different PY180-solvent samples. Moreover, there is no difference in the scattering characteristics of the Raman spectra after pigmentation [3,32,33]. From the Raman and IR spectra, it can be deduced that the pigmentation process does not change the molecular chemical bonding or generate new substances.

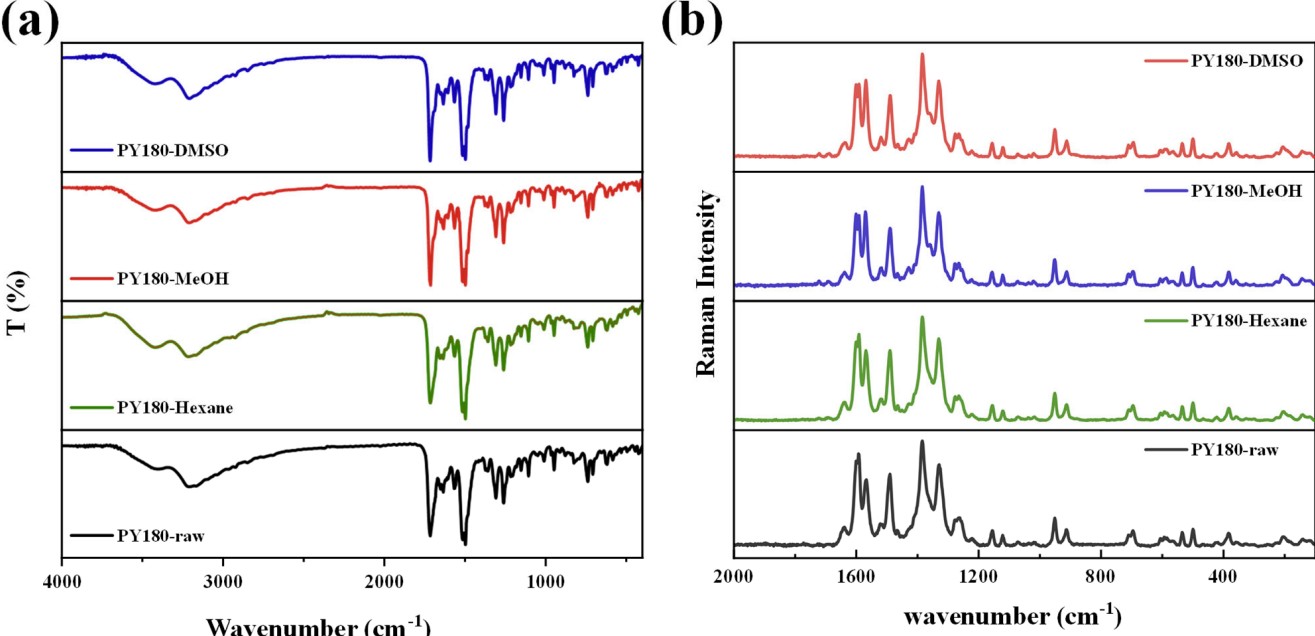

**Figure 2.** The FTIR (**a**) and Raman spectra (**b**) for PY180-raw, PY180-Hexane, PY180-MeOH, and PY180-DMSO.

XRD shows the crystallinity and crystal size and can be used to characterize the polymorphic form and determine if a crystalline transition has occurred. Many pigments are polymorphic (existing in different phases) with the same chemical composition but different molecular arrangements. The crystal structure has an impact on the chemical and physical properties, causing variation in color among phases. The PY180 diffraction peaks obtained in different solvents appeared at the same position in Figure 3a, indicating that almost the same phase is formed in all the samples. The number and position of the diffraction peaks did not change after the treatment, demonstrating that no crystal phase transformation occurred. The different heights of the peaks are related to the different orientations of each crystalline plane. When the crystallinity has increased, the intensity of the diffraction peaks becomes more pronounced. Among them, PY180-DMSO has the largest crystallinity, and hexane has the smallest.

The diffraction patterns of PY180-MeOH at different pigmentation times were monitored to study the pattern between crystallinity and color variation. Figure 3b shows the diffraction patterns of PY180-MeOH collected at 0, 1, 2, 4, 6, 8 and 10 h, respectively. Despite the similarities, there is a significant evolution in the XRD along the pigmentation time. This indicates that the pigmentation process provoked an important enhancement in

the crystallinity of the samples. The process could be divided into two phases, especially in MEOH solvent, where the main reflection evolves progressively from almost amorphous peaks to distinct diffraction peaks within 2 h of treatment time, indicating a considerable increase in crystallinity. However, in the second stage, the subsequent trend in the enhancement of the diffraction peak intensity slows down with prolonged pigmentation time. This may be due to the small crystal size with high interfacial energy in the early stages, which leads to a high rate of ripening. In the second stage, the low surface interfacial energy of large particles causes a lower ripening rate. Those trends observed therein are in good agreement with the changes in the optical behavior of the pigments, indicating that crystallinity is an important influencing factor for the color of PY180. This also proves that the solvent used for the Ostwald ripening process plays an important role because the solubility of the initially formed small particles in this solvent determines the Ostwald ripening rate.

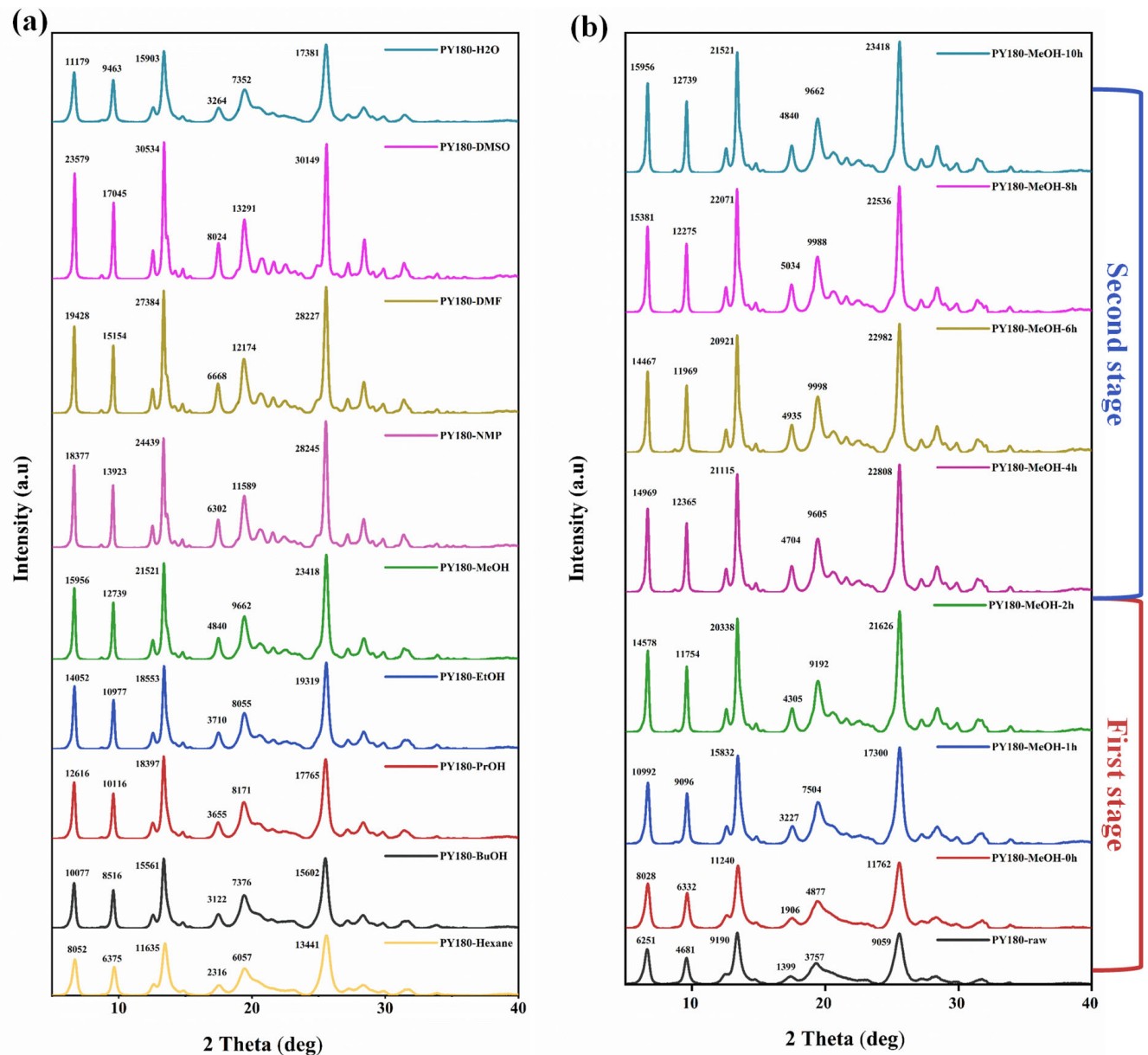

**Figure 3.** Experimentally observed PXRD patterns of (**a**) PY180 pigmented in the different solvents with the pigmentation time of 10 h, (**b**) PY180 in MeOH sample collected at different times.

To further study the variation in surface morphology of the particles, SEM images of the pigments pigmented by different solvents are shown in Figure 4. It shows that the untreated pigments are agglomerations of small-sized irregular, needle-like, and flaky particles. After treating with different polar hydrogen-bonded solvents, PY180 mainly presents a rod-like shape. In the presence of *n*-pentane, PY180 shows a granular shape with a small size and indistinct edges. Using polar non-protonic solvents, PY180 shows flakes with thickness and large size, especially in DMSO.

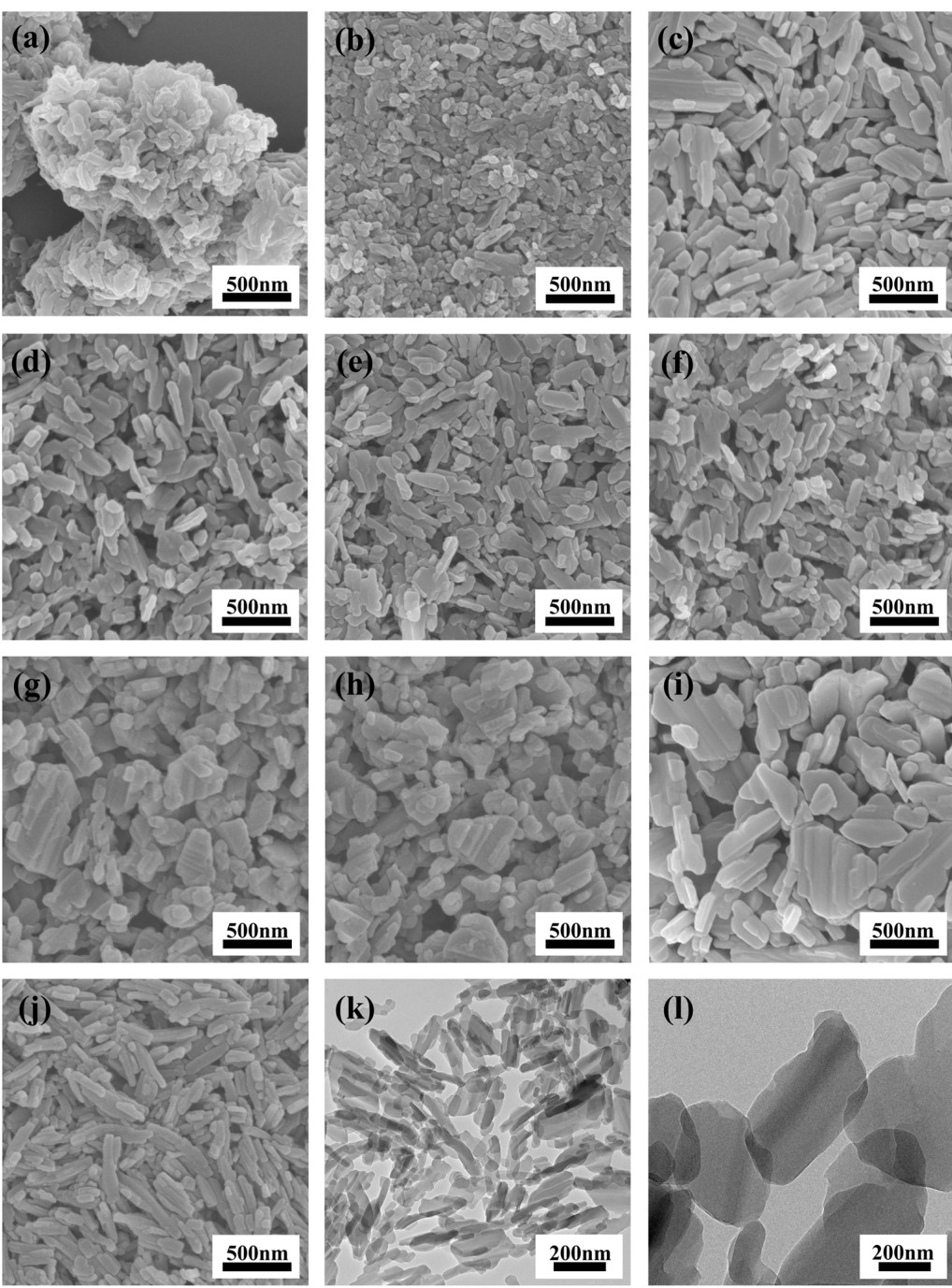

**Figure 4.** SEM images of PY180 (**a**) before and after treatment in different solvents: (**b**) hexane, (**c**) MeOH, (**d**) EtOH, (**e**) PrOH, (**f**) BuOH, (**g**) NMP, (**h**) DMF, (**i**) DMSO, (**j**) H$_2$O. TEM images of (**k**) PY180-MeOH, (**l**) PY180-DMSO.

When the particles are small, their high surface energy will lead to agglomeration and a dark color effect. Examples include PY180-Hexane with a small particle size and PY180-$H_2O$ with a large aspect ratio. The higher L-values of the pigments after treatments in DMSO NMP and DMF may be due to the low surface energy of the large-sized particles and improved dispersion between the particles. This enhanced dispersion facilitates the reflection and refraction of light.

TEM characterizations of PY180-MEOH and PY180-DMSO were performed, confirming that PY180-MEOH mainly appeared as rods and PY180-DMSO as flakes, where the PY180-DMSO flakes tend to grow as rods fused rather than aggregated, possibly due to the nondirectional attachment.

### 3.3. Effects of Pigment Solubilities in Different Solvents

From the above studies, it is evident that the solvent remains a significant influencing factor. The pigmentation process can be referred to as a recrystallization process, which is a thermodynamically driven dissolution–reprecipitation phenomenon. However, solubility is challenging to estimate for low-soluble systems. By the basic thermodynamic rules of the Gibbs phase, there is no particular difference between dispersion and dissolution. Then, we postulate that evaluating dispersion conditions could serve as a criterion for solvent selection in the recrystallization process.

Solubility parameters are commonly used to assess the compatibility of two substances and select appropriate solvents for polymers. Additionally, they also have direct relevance to the suspension and settling behavior of pigments in solvent mixtures. In this study, we employed the dispersion situation to determine the solubility parameters of PY 180, as it exhibits poor solubility in the solvent. We selected a range of polar and nonpolar solvents to investigate the dispersibility of PY180 in different solvents. It can be seen from Figure 5a that after centrifugation, the dispersibility can be easily assessed by observing the color of PY180 being dispersed in solvents. Our results clearly show that, among these ten solvents, the color is most pronounced in DMF, NMP, DMSO, and ACN, while it is almost colorless in water, n-Hexane, and methanol.

The difference between PY180 and different solvents was subsequently determined according to the UV spectra. Since the concentrations and absorption coefficients of the solvents could not be determined, the wavelengths of the most dominant visible peaks (A/I) were used instead of concentration. The A/I and solubility parameters are shown in Figure 5b–d. Then, the $\Delta\delta$ values of PY180 can be calculated in these solvents based on the $\delta D$, $\delta P$, and $\delta H$ values of each solvent as well as PY180 itself (Table S1). According to the rule of "like dissolves like", the individual forces of the three-dimensional solubility parameters are now discussed separately for solvents with a high dispersion of PY180 with $\delta d$, $\delta p$, and $\delta h$ in the range of 15.3–18.4 MPa$^{0.5}$, 12.3–18 MPa$^{0.5}$, and 6.1–11.3 MPa$^{0.5}$, and an optimal equilibrium value exists. Therefore, solutions with high polarity and moderate hydrogen bonding ability are required to obtain a good PY180 dispersion solution.

From the deduced solubility parameters mentioned earlier, it can be determined that solvents with good dispersion effects for PY180, such as DMSO, NMP, and DMF, play a positive role in the pigmentation process. Solvents with poor dispersion effects, like hexane and $H_2O$, are not good solvents for the pigmentation process. However, this conclusion exhibits a different phenomenon in alcohol solvents. This may be due to the fact that it primarily examines the intermolecular forces between the surface of particles and the solute molecules.

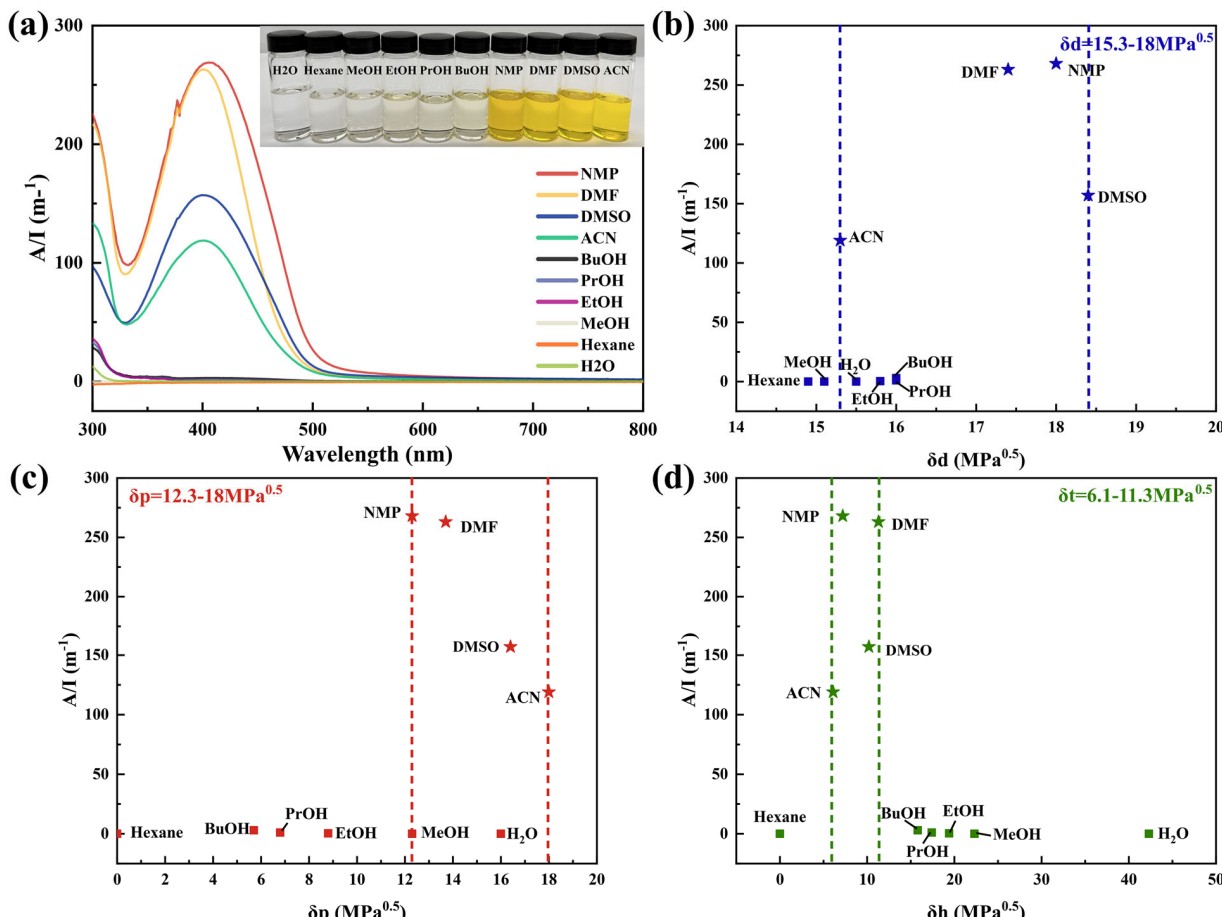

**Figure 5.** (**a**) Photographs and UV/Vis absorbance spectra (solutions diluted by a factor of 2) of PY180-raw suspensions in solvents after centrifugation. The plot shows absorbance per cell path length (A/l) at 400 nm of PY180/solvent and their correlation to Hansen solubility parameters of different solvents: (**b**) dispersive, (**c**) polar, and (**d**) hydrogen bonding. The graph displays data for "good" solvents, represented by star-shaped data points.

### 3.4. Solvent Effects Quantified Using Polarity Index

The solubility parameters obtained based on solid–liquid dispersion give some general influence rules, as low $\Delta\delta$ induces high $\Delta E$. However, there are some deviations, e.g., PY180-raw is poorly dispersed in methanol, while the color of PY180-MEOH is close to that of PY180-NMP. Interestingly, the resemblance between MeOH and NMP is the similar polarity. In this work, we explored this phenomenon from the perspective of molecular polarity.

The presence of distinct atoms and groups situated on PY180 can considerably influence the overall electron cloud of the molecule. To ascertain the conceivable electrostatic interactions between the molecule and its surroundings, we conducted an ESP analysis on its vdW surface following geometry optimization. This analysis is also indicative of intermolecular electrostatic interaction energy. From Figure 6a, it can be seen that electrostatic potential distribution on the surface of PY180 is not homogeneous and distributes over a relatively wide range (−40 to 50 kcal/mol). Among them, the lower ESP values are mainly represented by benzene rings and alkanes (−15 to 15 kcal/mol) and the vdW surface with a large negative value of ESP attributed to the surface close to the oxygens of the carboxyl group. The positive value is provided mainly by the hydrogen of the amino groups. Maxima and minima of ESP on the surface are −38.32 and +48.01 kcal/mol, stemming from the oxygen and hydrogen atoms of the benzimidazole moiety, respectively, which are prone to be a site of electrostatic interactions.

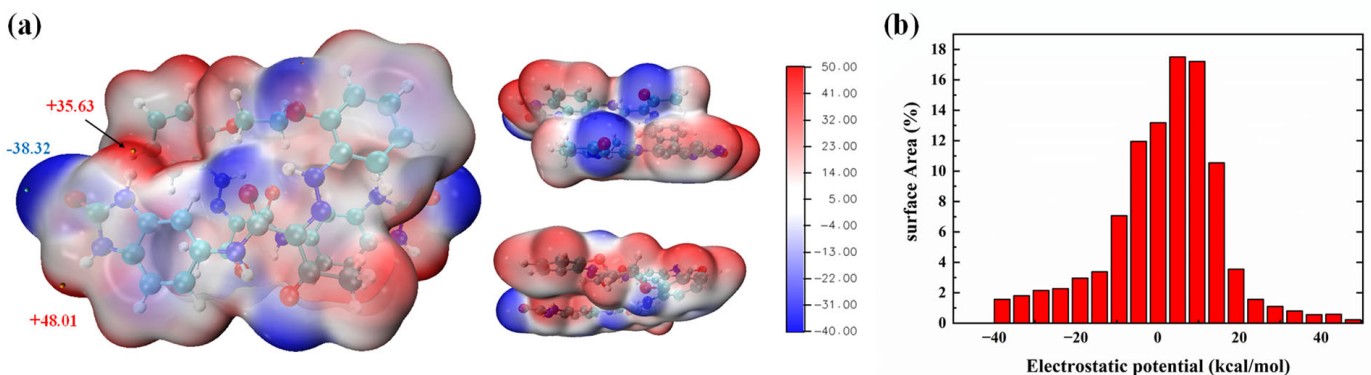

**Figure 6.** (**a**) ESP mapped molecular vdW surface of PY180 and (**b**) area percent in each ESP range. (B3LYP-D3/Def2-QZVP level). C—grey, H—white, O—red, N—blue. Blue area for negative ESP, red area for positive ESP. The extreme points of MEP points (kcal mol$^{-1}$) are highlighted.

The molecular polarity index (MPI) has been proposed to show the relationship between electrostatic potential distribution and polarity, which is expressed as [30]:

$$\text{MPI} = \left(\frac{1}{A}\right) \iint_S V(r)|dS$$

where V (r) refers to the ESP value at point r in space, and the integration is over the entire molecular surface S; A to the area of vdW surface. The MPI of PY180 was calculated as a value of 10.91 from Multifwn 3.8 dev software, indicating that PY180 is a polar molecule. Its intermolecular interaction force with polar solvents will be significantly greater than that of non-polar solvents.

The polarity index primarily represents the contribution to the electrostatic force or dipole–dipole interaction force. The larger its value, the greater the deviation from zero of the average electrostatic potential at the surface of the molecule. Therefore, the greater the polarity, the greater the overall ability to bond with other substances. The MPI values for all solvents were calculated from the ESP based on the surface of the solvent molecules using Multifwn 3.8 dev software, as detailed in Table 2. As shown in Figure 7, the effects of HSPs and MPI were systematically compared to investigate which factor plays a key role in the pigmentation process. The color changes after the pigmentation process with different solvents were ranked according to Δδ from large to small in Figure 7a. It can be found that the solvent molecules are similar to the HSPs of PY180, i.e., solvents with medium hydrogen bonding parameters and high polarity parameters would favor the recrystallization of PY180 during the pigmentation process, while solvents with high Δδ lead to low ΔE of pigments. Figure 7b shows a positive correlation between the MPI of solvents and color variation. This result suggests that color change is more obvious with a larger MPI of solvent, which means that the dipole–dipole interaction generated by polarity takes an important role. For instance, in alcohol solvents, the high polarity of MeOH facilitates pigment conversion more effectively compared to the relatively lower polarity of BuOH.

**Table 2.** MPI of solvent molecules (B3LYP-D3/Def2-QZVPlevel).

|  | Hexane | BuOH | PrOH | EtOH | MeOH | NMP | DMF | DMSO | H$_2$O |
|---|---|---|---|---|---|---|---|---|---|
| MPI | 2.835 | 9.043 | 9.933 | 11.167 | 13.224 | 14.025 | 15.830 | 17.774 | 21.244 |

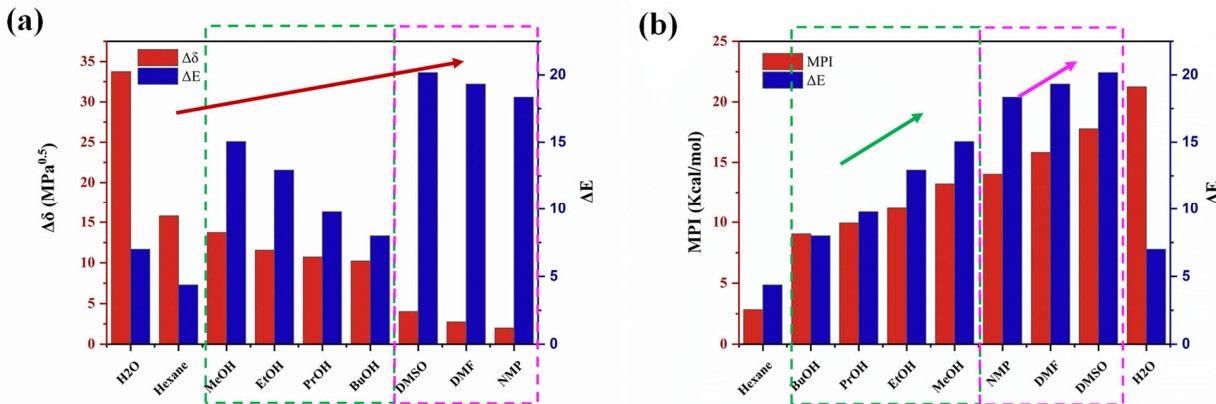

**Figure 7.** (**a**) Difference in solubility parameter (Δδ in red), (**b**) molecule polarity index (MPI in red) versus ΔE (blue). Green dashed lines for polar hydrogen solvent, magenta dashed line for aprotic polar media. The arrows indicate the overall trend of ΔE.

Consequently, HSPs can be used for rapid solvent screening and preliminary assessment making. MPI, on the other hand, is based on molecular polarity and intermolecular forces that can assist us more in understanding the mechanism of the pigmentation process. So, MPI can be utilized to supplement and correct HSPs for more accurate prediction.

## 4. Conclusions

In this study, the chromaticity changing trend of PY180 in the pigmentation process of different solvents was investigated, and the solvent's effect mechanism was deduced by a method that combined HSP theory with MPI analysis.

The chromaticity results indicate that the brightness of PY180 has been enhanced by the pigmentation process and shifted its color from red to green. IR, Raman, SEM, and XRD measurements indicate that there is no change in the chemical and crystal structure of PY180 after the pigmentation processes. The variation in color can be mainly attributed to the higher crystallinity and crystal size. Moreover, the change of XRD-MeOH diffraction peaks with time proves that pigmentation is a surface energy-driven process. When the surface energy decreases in value, the low recrystallization rate will lead to a slower color change.

The solubility parameter and the molecular polarity index were combined to explain the solvent effect on the dissolution–reprecipitation behavior of PY180. According to the HSP theory, it was found that the solvents with a small solubility difference from PY180 would tend to induce a significant change in chromaticity. The dispersion model mainly considered the interaction between a significant number of surface atoms and the solvent, which lacks precision. The results of the computer simulation indicate some relevance in the MPI and ΔE values. This could be attributed to the dipole–dipole interactions, which may play a dominant role between solvent and solute molecules. Solvents with higher polarity are more favorable for solute–solvent interactions. Under these interactions, the solute dissolution in the solvent and the subsequent rearrangement and reprecipitation process are promoted, thereby enhancing the rate of color transformation. This paper mainly discusses solvent effects based on the polarity aspect; other computational methods should be considered in the future to enrich the study from multiple perspectives.

**Supplementary Materials:** The supporting information can be downloaded at https://www.mdpi.com/article/10.3390/pr11102951/s1. Refs. [16,17,34] are cited in the Supplementary Materials.

**Author Contributions:** Methodology, K.Y.; writing—original draft, K.Y.; funding acquisition, Y.Y. and H.C.; supervision, L.D.; writing—review and editing, H.W. and L.D.; data curation, J.W. All authors have read and agreed to the published version of the manuscript.

**Funding:** This research was funded by of Anshan Hifichem Co., Ltd., China.

**Data Availability Statement:** The data that support the findings of this study are available from the corresponding author upon reasonable request.

**Acknowledgments:** The authors are grateful for the financial support of Anshan Hifichem Co., Ltd., China.

**Conflicts of Interest:** The authors declare no conflict of interest.

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
