# Peer review of "Investigation into Pigmentation Behaviors and Mechanism of Pigment Yellow 180 in Different Solvents"

_processes, doi:10.3390/pr11102951_

Round 1

Reviewer 1 Report

Respected authors.

Please correct the following aspects to improve the quality of the manuscript:

Modify the paragraphs between lines 58-61, 109-112, 242-245, and 270-273 to reduce coincidence with bibliographic sources.

The manuscript is very well written. The pigmentation behavior of PY180 and its stability in various solvents is well described. The methods used, such as chromaticity analysis, SEM, and solubility studies, are adequately described, and their results are correctly analyzed, demonstrating a contribution to the chemical stability of PY180 with a view to its potential use. I suggest including a second computational method, such as HF, to compare the calculated parameters. It could reinforce the analysis of results.

Due to the above, I consider that the manuscript should be published.

Author Response

Thank you very much for your thoughtful suggestions and insights. The manuscript has benefited from these insightful suggestions. Please see the attachment.

Reviewer 2 Report

The manuscript "Probe in Pigmentation Behaviors and Mechanism of Pigment Yellow 180 at different solvents" by Kairu Ye et al. describes the change in color of Pigment Yellow 180 under the action of various solvents. The manuscript is ill-written, and I do not recommend this manuscript for publication.

Comments:

1) This work's motivation and aim are unclear to the reader.

2) English is wrong, and the text should be heavily rewritten.

3) The theoretical DFT calculations do not explain anything in this work and look redundant.

4) The description of theoretical DFT calculations in Section 2.2. has no sense.

5) IR spectra of a sample treated by hexane contain signals of hexane itself about 2900 cm-1.

English is bad, and the text should be heavily rewritten.

Author Response

(The authors gave the same response as above.)

Reviewer 3 Report

The article requires following improvements:

1. Please improve image quality for all the figures. They are unreadable.

2. The authors mentioned "From the Raman and IR spectra, it can be deduced that the pigmentation process does not change the molecular chemical bonding or generate new substances." Figure 2 (a) depicts the FTIR spectra for PY180-Hexane which has an additional peak near 2800-2900 cm-1 wavenumber. Please explain the peak.

There are a couple of minor grammatical errors in the manuscript. Please correct those. Please take professional help, if needed.

Author Response

(The authors gave the same response as above.)

Round 2

Reviewer 2 Report

The authors have improved the manuscript.

The authors have improved the manuscript.